# Gut Microbiome in Inflammatory Bowel Disease: Role in Pathogenesis, Dietary Modulation, and Colitis-Associated Colon Cancer

**DOI:** 10.3390/microorganisms10071371

**Published:** 2022-07-07

**Authors:** John Gubatan, Theresa Louise Boye, Michelle Temby, Raoul S. Sojwal, Derek R. Holman, Sidhartha R. Sinha, Stephan R. Rogalla, Ole Haagen Nielsen

**Affiliations:** 1Division of Gastroenterology and Hepatology, Stanford University School of Medicine, Stanford, CA 94305, USA; mtemby@stanford.edu (M.T.); rsojwal@stanford.edu (R.S.S.); drholman@stanford.edu (D.R.H.); sidsinha@stanford.edu (S.R.S.); srogalla@stanford.edu (S.R.R.); 2Department of Gastroenterology, Medical Section, Herlev Hospital, University of Copenhagen, DK-2730 Copenhagen, Denmark; theresa.louise.boye@regionh.dk (T.L.B.); or ole.haagen.nielsen@regionh.dk (O.H.N.)

**Keywords:** gut microbiome, inflammatory bowel disease, ulcerative colitis, Crohn’s disease, diet, colon cancer, immunity

## Abstract

The gut microbiome has increasingly been recognized as a critical and central factor in inflammatory bowel disease (IBD). Here, we review specific microorganisms that have been suggested to play a role in the pathogenesis of IBD and the current state of fecal microbial transplants as a therapeutic strategy in IBD. We discuss specific nutritional and dietary interventions in IBD and their effects on gut microbiota composition. Finally, we examine the role and mechanisms of the gut microbiome in mediating colitis-associated colon cancer.

## 1. Microbiome in the Pathogenesis of Inflammatory Bowel Disease (IBD)

The individual human lives in symbiosis with 100 trillion microbiota of the gastrointestinal tract, comprising more than 1000 different types, which are distributed among the genera; bacteria, bacteriophages (bacterial viruses), fungi, and protozoa [1]. The fungal and protozoan microbiome implications in IBD are, however, poorly described [2]. Sparse studies suggest that the protozoan microbiome in patients with active IBD display an increased prevalence of Blastocystis compared to those with quiescent disease or control subjects [3,4]. Additionally, studies report alterations in the diversity (measure of the number of species in a community, and a measure of the abundance of each species) and composition of the fungal microbiome among patients with IBD compared to healthy subjects [5,6,7,8,9,10], and, moreover, intra-individual changes in the fungal composition between inflamed and noninflamed mucosa have been observed in patients with Crohn’s disease (CD) [11]. Intestinal bacteria with associated bacteriophages and the intestinal epithelial cell layer are increasingly being studied, and exist in a dynamic tripartite—both mutualistic and parasitic—relationship, which recently started to be unraveled (Figure 1). Pattern recognition receptors (PRRs) specialized in recognizing bacteria and bacterial products are found in both immune and intestinal epithelial cells. In this way, intestinal epithelial cells balance the composition and luminal microbiota by regulating the secretion of mucus, antimicrobial peptides, and immune mediators, e.g., mucosal immunoglobulin A (IgA) [12]. Nevertheless, surprising evidence also points towards direct communication between bacteriophages and intestinal epithelial cells by bacteriophages adhering to mucosal surfaces, apical-to-basolateral transcytosis (i.e., endocytosis followed by exocytosis transporting bacteriophages across epithelial cells), and by the direct delivery of proteins and nucleic acids to eukaryotic cells [13]. In the following, we will focus on the bacteria and bacteriophages with respect to the pathogenesis of IBD.

### 1.1. Bacteria in IBD

The microbiome of patients with IBD is characterized by bacterial dysbiosis (i.e., an imbalance of pathogenic and commensal bacteria). Bacterial diversity has been shown to be reduced during active inflammation in IBD [14,15]. Furthermore, gut microbiome composition has been shown to vary based on their location along the gastrointestinal tract [16]. This observation is probably driven by mucosal changes in tissue oxygenation and disruption of the mucosal barrier function in IBD [15]. Bacterial dysbiosis, which refers to an imbalance of pathogenic and commensal bacteria, is in IBD characterized by a depletion of the phyla *Actinobacteria*, *Firmicutes*, and *Bacteroidetes* [17,18,19,20], and an enrichment of *Proteobacteria* [21]. Interestingly, Firmicutes and Bacteroidetes are primary producers of energy substrates for intestinal epithelial cells and anti-inflammatory agents, including butyrate and other short-chain fatty acids (SCFAs) [22,23]. Not surprisingly, fecal samples of patients with IBD display a decreased amount of SCFAs [24]. Moreover, long-term remission normalizes both the bacterial microbiota and SCFAs levels in a majority of IBD patients, although with pronounced interindividual variations [25,26,27]. Additionally, low levels of *Firmicutes* and *Faecalibacterium* species appear to be related to a high risk of relapse and post-operative recurrence of IBD patients [28,29,30,31]. Polymorphisms of the *NOD2* gene are associated with an abundance of *Faecalibacterium prausnitzii*, the *Roseburia* genus and the *Enterobacteriaceae* family [32,33]. Additionally, the microbiome is affected by the diet of the host [34,35]. Interestingly, the intake of prebiotics such as nondigestible fibers is positively correlated with circulating serum levels of granulocyte-macrophage colony stimulating factor (GM-CSF) and negatively correlated with interleukin (IL)-6 and IL-8. These cytokines play central roles in the pathogenesis of IBD [36] and could be a result of altered bacteria or bacterial metabolites in the intestinal lumen. Thus, an intimate relationship between host bacterial microbiome and epithelial cells is evident in the pathogenesis of IBD. Hence, bacteria or bacterial products regulate components of the immune system, but an intestinal chronic low-grade inflammatory environment causing tissue oxygenation and disruption of the mucosal barrier may, on the other hand, significantly impact the microbiome by selecting against inflammatory sensitive species and inducing blooms in evolutionary adapted species.

### 1.2. Bacteriophages in IBD

The virome of the gut is dominated by viruses that infect bacteria, the so-called bacteriophages (phages), that can present themselves as RNA or both double- and single-stranded DNA [37]. Thus, patients with IBD display an elevated intestinal phage diversity and abundance [38,39]. Importantly, this expansion and diversification of the intestinal bacteriophages is not secondary to the observed concomitant and significantly reduced bacterial diversity [39]. 

Bacteriophages can indirectly stimulate the immune system by mediating bacterial lysis, which subsequently cause the release of phosphorus-containing bacterial components along with active enzymes [40], but they can also be directly sensed by intestinal epithelial cells and innate immune cells. In fact, bacteriophages have recently been found to be embedded within the intestinal mucus, and are transported across the intestinal epithelial barrier via transcytosis [13]. 

Moreover, a recent study has proposed a possible mechanism for bacteriophage-mediated mucosal immunity [41]. This murine study suggested that increased bacteriophage levels may exacerbate colitis via the nucleotide-sensing receptor, Toll-like receptor (TLR) 9, and IFN-γ on immune cells [41]. Together with a positive correlation between mucosal IFN-γ and bacteriophage levels in patients with ulcerative colitis (UC), IFN-γ was proposed to be important for bacteriophage-mediated mucosal immunity and IBD [41]. Taken together, the above-mentioned studies underscore the importance of understanding the direct effects on bacteriophages, not only on bacteria, but also on both immune and epithelial cells.

One of the major obstacles to comprehensively defining the virome is “viral dark matter”, i.e., metagenomic sequences originating from viruses, which, do not align with any reference virus sequences [42]. This is caused by a lack of universal marker genes on phages (similar to the 16S ribosomal RNA gene in bacteria or the 18S and internal transcriptional spacer (ITS) ribosomal RNA genes in eukaryotes), a lack of taxonomic information due to poorly populated databases, and the fact that the virome exhibits an enormous diversity and interindividual variation [43]. Additionally, bacteriophages remain hard to culture and are challenging to analyze. Nevertheless, recent data using whole-virome analysis have shed some light on the viral dark matter in IBD [44]. Intestinal bacteriophages exist in two states: lytic or temperate. The lytic cycle results in destruction of the infected cells, and the temperate phages integrate their genomes into their host bacterial chromosome [45]. At some point, temperate bacteriophages can switch from the lysogenetic life cycle to the lytic life cycle. Interestingly, in this study, the temperate phage population displayed a shift from lysogenic to lytic replication in patients with IBD [44]. Unlike prior database-dependent methods, no changes were observed in viral richness (number of species in a community) in healthy subjects compared to patients with IBD [44], which challenges the current knowledge of a phage-related IBD pathogenesis. More research, with targeted analyses of the viral dark matter, is needed to unravel the nature of bacteriophage-mediated mucosal immunity in IBD.

**Figure 1 microorganisms-10-01371-f001:**
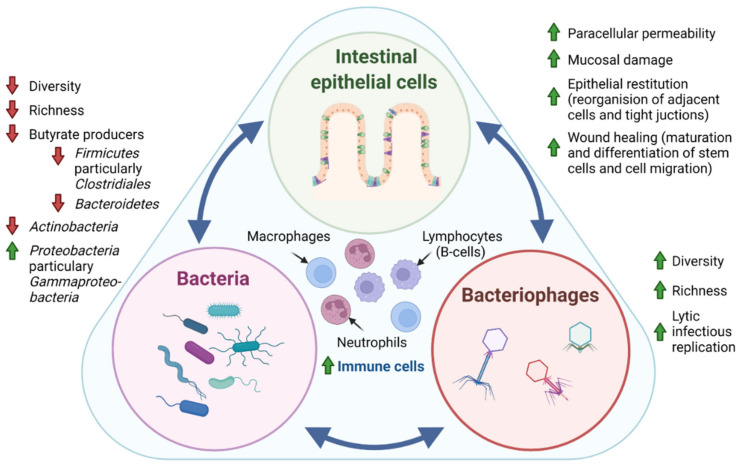
**Tripartite relationship between the intestinal epithelial cells, bacteria, and bacteriophages in IBD pathogenesis.** In IBD pathogenesis, bacterial dysbiosis is characterized by decreased bacterial diversity (measure of the number of species in a community, and a measure of the abundance of each species) and richness (number of species in a community) evident by the depletion of the phyla *Actinobacteria, Firmicutes*, and *Bacteroidetes* and an enrichment of *Proteobacteria*. In contrast, studies generally suggest that intestinal bacteriophages, which are viruses that infect and replicate within bacteria, display increased diversity and richness. Interestingly, it has recently been suggested that the temperate phage population displays a shift from lysogenic to lytic replication in patients with IBD [44]. Where intestinal epithelial cells are known to directly regulate the secretion of mucus, antimicrobial peptides, and immune mediators through patterns recognition receptors (PRR), surprising evidence also points towards direct communication between bacteriophages and epithelial cells by adhering to mucosal surfaces, apical-to-basolateral transcytosis, and by the direct delivery of proteins and nucleic acids to eukaryotic cells. Thus, the intestinal epithelial cell layer, intestinal bacteria, and bacteriophages exist in a dynamic tripartite—both mutualistic and parasitic—relationship. Further, sparse studies propose that fungal and protozoan microbiomes are also affected in IBD pathogenesis, displaying both altered diversity and composition. The mechanistic interplay between intestinal epithelial cells, bacteria, bacteriophages, as well as fungi and protozoa, has yet to be unraveled, but would potentially provide insight for future clinical applications of microbiota in IBD. Green arrow: increased, red arrow: decreased.

### 1.3. Clinical Relevance of Gut Microbiota in IBD

Previously, it was believed that patients with CD would benefit from antibiotic therapies, resulting in a deleterious effect on the intestinal microbiota [46]. Nonetheless, exposure with antibiotics has been associated with increased microbial dysbiosis [47], and no scientific evidence exists for a beneficial effect of the antibiotic treatment of patients with CD without fistulas or ongoing infections. Instead, an increasing number of clinical trials have been initiated with the aim of investigating the therapeutic effect of fecal microbiota transplantation (FMT) in patients with IBD [48,49]. 

In the largest, double-blind, randomized, placebo-controlled clinical trial of donor FMT for UC to date, the primary outcome was defined as steroid-free clinical remission with endoscopic remission or response. The primary outcome was achieved in 11 (27%) of 41 patients allocated FMT versus three (8%) of 40 who were assigned placebo (*p* < 0.04) [50]. Another study of adults with mild to moderate UC compared anaerobically prepared pooled donor FMT versus autologous FMT. Here, 12 of the 38 participants (32%) receiving pooled donor FMT, as compared to 3 of the 35 (9%) receiving autologous FMT, experienced an 8-week steroid-free clinical remission (*p* < 0.03) [51]. These studies and other published data indicate that donor FMT induces remission in a statistically significant proportion of UC patients [50,51,52,53]. 

However, available studies in patients with CD are scarce and under-powered. One study with adult colonic or ileo-colonic CD (*n* = 17, 8 FMT and 9 sham) showed that the steroid-free clinical remission rate at 10 and 24 weeks was 4 of 9 (44%) and 3 of 9 (33%), respectively, in the sham transplantation group and 7 of 8 (88%) and 4 of 8 (50%) in the FMT group (*p* > 0.05 at both time points), and none of the patients reached the primary endpoint [54]. These results are currently being tested in a larger ongoing clinical trial (*n* = 24) (ClinicalTrials.gov identifier NCT02097797). Interestingly, several studies in both UC and CD have revealed a significant shift in fecal microbial composition towards a greater microbial diversity, like that of healthy subjects in patients who experienced clinical responses [53,55,56,57]. Although FMT studies to date report low FMT-associated adverse effects in UC [52,56], one study reported flares within a few days of undergoing FMT in CD [57]. Thus, further research is warranted to assess the long-term maintenance of remission and safety of both donor and therapeutically optimized donor-derived strains [58]. Other important issues to address include the applied delivery method (e.g., delivery via colonic transendoscopic tube or encapsulated delivery either orally or via topical application [59,60]); fecal sample preparation and storing (fresh or frozen), or if one should help facilitate the colonization of microbiota using bowel lavage or antibiotics, risking the elimination of preexisting beneficial bacteria prior to the therapeutic intervention. Additionally, in the future, microbiota may possibly be used as a noninvasive biomarker for preventative, diagnostic, prognostic, and monitoring purposes among patients with IBD [61,62].

## 2. Dietary Modulation of the Gut Microbiome in IBD

Interests in the diet’s ability to alter the gut microbiome as a therapeutic strategy among patients with IBD has grown tremendously in recent years (Figure 2) [63]. Several nutritional therapies have been explored among pediatric patients with IBD. For example, the Crohn’s Disease Exclusion Diet (CDED) is a high protein, low-fat diet that includes foods such as chicken, fish, eggs, rice, potatoes, and various fruits and vegetables. This intervention has been effective for mild to moderate CD in children, as well as for patients whose response to anti-TNF biologic treatments plateaued [63]. Levine et al. found that remission in the CDED groups was associated with changes in microbial diversity, a decrease in *Proteobacteria* and an increase in *Firmicutes*, particularly *Clostridiales*. Remission also led to a significant decrease in *Proteobacteria*, particularly *Gammaproteobacteria* [64].

A Ketogenic Diet (KD) and Low-Carbohydrate Diet (LCD) show promising changes in the specific composition and function of gut microbiota and metabolites in mice [65]. The study by Kong et al. found that, after inducing colitis, KD significantly reduced inflammatory responses, protected intestinal barrier function, and reduced the expression of inflammatory cytokines, whereas the opposite effects were observed for the LCD [65]. These findings indicate a promising dietary strategy for treating IBD, and demonstrate, for the first time, that fecal microbiota transplantation from donors on a KD confers microbiota benefits and relieves colitis in dextran sulfate sodium (DSS)-induced recipients [64]. KD dramatically increased the abundance of *Akkermansia* and *Roseburia*; expanding the abundance of *Akkermansia* has been associated with improved glucose homeostasis, modulated immune responses, and protected barrier function [65]. It should, however, be noted that while KD alleviated the progression of intestinal inflammation, it also reduced the abundance of some healthy bacteria, such as *Lactobacillus*, compared with a normal diet [65]. 

Fiber in fruits and vegetables has been shown to provide several benefits to patients with IBD, such as prolonging remission and reducing lesions in the intestinal mucosa, while an imbalance in the consumption of fiber is a risk factor for IBD development [66]. Furthermore, a diet rich in oats prevents the worsening of gastrointestinal symptoms in UC, while a diet rich in high-fiber legumes mitigates intestinal inflammation in rodent models of IBD [66]. 

As previously mentioned, the production of SCFAs from fiber in gut microbiota has been established as a protective agent against IBD in recent years [66]. The most-studied SCFAs (butyrate, acetate, and propionate) exert anti-inflammatory effects in IBD by inhibiting NF-ĸB activation to suppress cytokines [66]. Butyrate, produced by the microbe *Firmicutes,* exists at highest concentrations in the colon and cecum, and propionate and acetate are byproducts from *Bacteroidetes* in the small and large intestines. SCFA production from fiber is crucial for reducing the inflammatory response in patients with IBD; therefore, a diet rich in fiber is essential to gut health [66]. 

Alternatively, fish consumption can lower the risk of IBD. Studies have found ω3FAs to support anti-inflammatory processes when interacting with microbes and alter microbiota diversity, increase beneficial bacteria, and reduce harmful bacteria. ω3FAs encourage growth of SCFA-producing microbes, including the *Lachnospiraceae,* and lessen the abundance of pathogenic microbes, such as *Enterobacteriaceae,* in infants [66]. However, the exact pathways and interactions between ω3FAs and the microbes themselves remain unclear. 

As interest grows in the benefits of a plant-based diet in IBD, studies have demonstrated that processed and animal-derived foods, in contrast, are associated with higher abundances of CD and UC inflammatory species such as *Ruminococcus*, as well as with an elevated calprotectin, the gut-specific inflammatory marker [67]. Allin et al. found that processed meat, soft drinks, refined sweetened foods, and salty foods are associated with a higher risk of developing IBD [68]. The study associates the excessive ingestion of ultra-processed foods (UPF) with an increased risk of IBD. Thus, compared with one serving of UPF per day, 5 or more servings per day was associated with a hazard ratio of IBD of 1.82 (95% confidence interval, 1.22–2.72). Unprocessed foods, such as white meat, dairy, starch, fruit, vegetables, and legumes, were shown not to be associated with IBD, while fried foods were associated with a higher rate of both CD and UC. IBD development is not affected by individual food categories (meats, dairy, starch, and fruit and vegetables), suggesting that consuming overly processed foods may be a major factor in diet-related IBD development [69]. 

Why and how processed/animal-derived foods in the gut may cause inflammation is still unknown. However, it is suggested that the processed sugars, red meats, and saturated fats abundant in the Western diet drastically alter the tissue and barrier function of the intestines, which trigger an inflammatory response leading to an imbalance of the TH17/Treg axis [69]. It is currently unknown whether Western diets also lead to adverse IBD outcomes in patients with well-established CD or UC; thus, a gap exists, which future studies must investigate.

## 3. Gut Microbiome and Colitis-Associated Colon Cancer

Patients with IBD are at a higher risk of developing colon cancer [70,71,72]. The two main types of IBD both greatly increase the risk for colitis-associated cancer (CAC). Thus, UC increases a patient’s cumulative risk of developing CAC by up to 18–20%, while CD increases the cumulative risk by up to 8% after having the disease for 30 years [73]. Although the disease etiologies of both IBD and CAC are complex, growing evidence suggests that the microbiome may play a major role in CAC (Figure 3) [74,75,76]. 

The large surface of the digestive tract is constantly in contact with both commensal and pathogenic bacteria. The luminal surface is covered in mucus, which acts as the first line of defense against these pathogens [77]. In turn, pathogenic bacteria have evolved different mechanisms to cross the mucus, then bind to and proliferate on epithelial cells [78]. The host defense systems are subsequently activated through the innate immune system, such as antimicrobial peptides (AMPs), which are stimulated by microbiota metabolites [79]. Defensins are the most common AMP in the gut and are effective in controlling targeted bacteria by forming pores in their membrane [80]. Concurrently, epithelial cells and immune cells of the intestinal wall recognize these bacteria through pattern-recognition receptors (PRRs). One of the roles of these PRRs is to act as a bridge between the innate and adaptive immune system.

There are four major PRR classes: toll-like receptors (TLRs), nucleotide-binding oligomerization domain-like receptors (NLR), C-type lectin receptors (CLR), and RIG-1 like receptors (RLR) [77,80]. Many of these PRRs, such as TLRs, recognize pathogens from their pathogen-associated molecular patterns (PAMPS), as well as danger-associated molecular patterns (DAMPs) that come from stressed or damaged cells [81,82]. Signaling from TLR4 might act as a pivotal pathogen-activated tumor signal pathway in the development of CAC [83,84]. TLR4 binds to lipopolysaccharide (LPS) from gram-negative bacteria, such as *Fusobacterium nucleatum* and *Salmonella* [85,86]. These two pathogens were strongly associated with the development of CAC [87,88,89,90]. Once LPS is bound to TLR4, the receptor complex triggers intracellular signaling, resulting in the transcription of inflammatory cytokines [91]. These inflammatory cytokines include TNF-α, IL-6, IL-1, and type I interferons [92]. In intestinal epithelial cells, Toll-like receptor (TLR) 4 expression is relatively low; however, it is significantly upregulated during IBD development and CAC [93]. TLR4 knockout mice given DSS to induce IBD showed insufficient epithelial repair. TLR4 also plays a role in the proliferation of intestinal epithelial cells [94,95,96]. Therefore, the proliferation-promoting effect from TLR4 is required for resistance against inflammation-induced intestinal damage [97]. However, the upregulation and chronic activation of TLR4 could lead to the development of CAC because of this proliferation-promoting property. In addition, TLR4 may provide malignant cells with protection from apoptosis [97]. It has been found that mice with acute colitis and TLR4 knockout had increased intestinal epithelial cell apoptosis [98]. This effect is necessary during colonic inflammation to protect and repair injured epithelial cells, but can have negative outcomes due to the onset of tumorigenesis. TLR4 is an example of how the host immune response may lead to CAC; however, the next section will describe how the microbiome itself can directly induce CAC.

Another way that the gut microbiota is involved in the development of CAC is through the production of protein toxins with carcinogenic effects [99]. These carcinogenic effects occur when toxins either target DNA causing genomic instability (genotoxins) or alter the cellular signaling, stimulating proliferation and resistance to apoptosis (cytotoxins) [100]. Two major types of genotoxins that have the potential to cause DNA damage are cytolethal distending toxin (Cdt) and colibactin [101]. Cdts are released by at least 30 pathogenic gram-negative bacteria, including *Salmonella* [102]. As a heterotrimer, the only enzymatically active subunit is the CdtB subunit [103]. Once it has been transported to the nucleus at low doses, the CdtB subunit can cause DNA single-strand breaks (SSBs), whereas at high doses, it can cause double-stranded breaks (DSBs), activating the DNA damage response [104]. However, chronic exposure to sublethal doses of Cdt can impact the damage response, causing reduced damage detection and increased mutations [105]. Another genotoxin is colibactin, secreted by *Escherichia coli* (*E. coli*) strains with the phylogenetic group B2 [106]. Colibactin’s chemical structure and genotoxic mechanism have remained elusive, because it is produced in small quantities and is very unstable [107]. It is believed that, similar to Cdts, it causes double-strand breaks (DSBs), incomplete DNA repair, and chromosomal instability [108].

While cytotoxins do not directly interact with DNA and, thus, do not explicitly cause mutations, they can induce CAC through cellular signaling, which affects cellular proliferation and cell cycle checkpoints. For example, *Bacteroides fragilis* toxin (BFT) is produced by enterotoxigenic *Bacteroides fragilis* causing diarrhea and epithelial damage [105]. BFT can lead to cleavage of the tumor suppressor, E-cadherin. The extracellular domain of E-cadherin is necessary for cell–cell contact and cell proliferation, while the intracellular domain is bound to β-catenin. Once dissociated, β-catenin becomes a transcription factor for cell proliferation. BFT also delays the apoptosis of intestinal epithelial cells. 

In CRC (patients alive five years after diagnosis) patients, there is an accumulation of pathogenic bacteria with a decrease in butyrate-producing bacteria [105]. Butyrate is a short-chain fatty acid (SCFA) that plays an important role in gut homeostasis by reducing the pH and oxygen levels, creating a favorable environment for anaerobic bacteria, and reducing the Enterobacteriaceae pathogens. Therefore, its reduction, combined with an increase in CAC-inducing bacteria, contributes to the development of CAC. The gut microbiome of CRC patients, when compared to healthy patients, has been characterized by an increase in *Fusobacterium* (particularly *Fusobacterium nucleatum*), *Enterococcus*, *Esherichia*/*Shigella*, *Bacteroides fragilis*, *Klebsiella*, *Peptostreptococcus*, and *Streptococcus* with a concurrent decrease in *Lachnospiraceae*, a butyrate-producing family of bacteria. 

As chronic colitis increases the chance of developing colon cancer, certain IBD therapies such as mesalamine (5-ASA) have been shown to have chemopreventive effects for CAC in observational studies [106,107]. It was discovered that 5-ASA has beneficial effects in UC patients by re-establishing a healthy gut microbiota. Dai et al. observed, through 16S rRNA sequencing, that there was a significant change in the gut microbiota of treatment-naïve UC patients [107]. There was an increase in the following genera: *Escherichia-shigella*, *Megamonas*, *Clostridium_sensu_stricto_1*, *Enterococcus* and *Citrobacter*. After 5-ASA treatment, 49 candidate genera were significantly reversed, including *Enterococcus*. *Enterococcus* is of particular interest because it was significantly correlated with UC pathogenesis [107]. In addition, *Enterococcus faecalis* has been shown to play a role in the development of adenocarcinoma in IBD. However, the mechanisms by which *Enterococcus* may cause CAC remain unclear [108].

Vitamin D deficiency is common among patients with IBD. It is appreciated that vitamin D may modulate intestinal immunity and suppress inflammation [109,110,111,112]. Studies in mice have shown that vitamin D can decrease CD4+ and CD8+ proliferation and subsequent inflammatory cytokines [113,114]. Furthermore, vitamin D may also alter inflammation by interacting with dendritic cells, macrophages, antigen-presenting cells, and NK cells [114]. Prior studies in mice have shown that vitamin D may reduce CAC. Murine models of colitis that were provided with supplemental vitamin D experienced a significant decrease in colon tumor formation, which was mediated through MAPK signaling [115]. In another study, conditional intestinal vitamin D receptor (VDR) knockout led to an increased number of colon tumors in a murine colitis, which shifted the gut bacteria profile to be more susceptible to carcinogenesis, as well as increasing secondary bile acids [116].

Finally, it is important to look at the role that vitamin D plays in altering the microbiome and how this may affect CAC. For example, in a study with pre-diabetic individuals who were vitamin-D-deficient, it was found that vitamin D supplementation was inversely correlated with *Firmicutes* (genus *Ruminococcus*) [117], one of the genera that was positively correlated with tumor counts in murine models of colitis [104]. Furthermore, vitamin D appears to stimulate the expression of PRRs, which could help protect the epithelial tissue layer in the colon from bacterial invasion [118]. In a prior study [119], Singh et al. found that, among vitamin-D-deficient patients, the gut microbiome between vitamin D supplementation responders versus non-responders showed significant differences in the major gut bacterial phyla. In a randomized, double-blinded study of vitamin D supplementation of healthy adults, increased concentrations of serum vitamin D were associated with an increased number of beneficial bacteria, and a decreased level of pathogenic bacteria [120]. Future studies, however, are needed to understand the chemoprotective effects of vitamin D on risk of colon cancer among patients with IBD and the mechanisms of how the gut microbiome may mediate this effect.

## 4. Conclusions

IBD is characterized by gut microbial dysbiosis with decreased beneficial/commensal bacteria (*Firmicutes*, *Actinobacteria*, and *Bacteroides*) and an enrichment of pathogenic/colitogenic *Proteobacteria*. There is increasing evidence that gut viruses, such as bacteriophages, may impact IBD, but studies have been limited by technological challenges with measuring and interpreting the gut virome. Fecal transplants have been explored in relatively small studies in patients with IBD, with mixed overall results (some benefits in UC, inconclusive results for CD). Dietary interventions, such as the CDED, ketogenic diet, low-carbohydrate diet, and plant-based diets, may have beneficial effects on patients with IBD by improving gut microbial diversity and increasing beneficial gut bacteria. Gut microbiota-derived antigens and toxins may mediate their carcinogenic effects by activating innate immune signaling pathways, stimulating cell proliferation/inhibiting apoptosis, and directly damaging DNA. The chemoprotective effects of mesalamine, as well as vitamin D and the VDR, on the development of colitis-associated colon cancer may be mediated through alterations in the gut microbiome. Future studies are needed to investigate whether manipulating the gut microbiome through these chemoprotective agents or other strategies may mitigate the risk of colitis-associated colon cancer among patients with IBD.

## Figures and Tables

**Figure 2 microorganisms-10-01371-f002:**
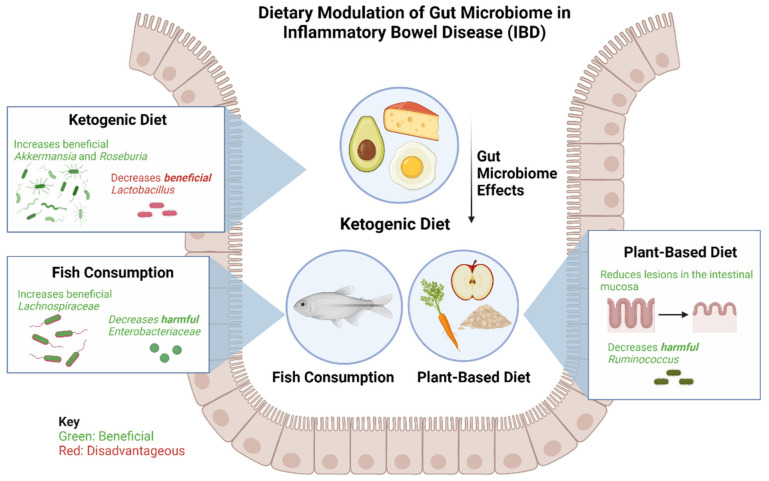
**Effects of a ketogenic diet, plant-based diet, and fish consumption on gut microbiome in patients with IBD.** The ketogenic diet has been shown to increase beneficial bacteria *Akkermansia* and *Roseburia* and consequently decrease beneficial *Lactobacillus*. The plant-based diet has been found to be beneficial in reducing lesions of the intestinal mucosa and reducing harmful *Ruminococcus*. Fish consumption leads to an increase in beneficial *Lachnospiraceae* and a decrease in harmful *Enterobacteriaceae*.

**Figure 3 microorganisms-10-01371-f003:**
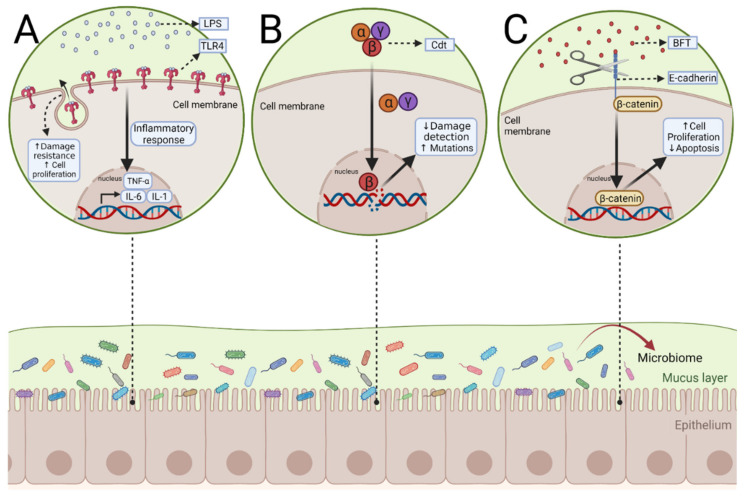
**Potential mechanisms of microbial activation of pathways leading to colitis-associated colon cancer.** (**A**) Lipopolysaccharides (LPS) produced by gram-negative bacteria such as *F. nucleatum* and *Salmonella* bind to the receptor TLR4. This leads to the transcription of inflammatory cytokines such as tumor necrosis factor (TNF-α), interleukin-6 (IL-6), IL-1, and type I interferons. During IBD, TLR4 is upregulated and may cause CAC, due to its proliferation-promoting ability. (**B**) In this scenario, the heterotrimer produced by pathogenic gram-negative bacteria, Cytolethal distending toxin (Cdt), can directly induce CAC. CdtB is the only active subunit and can make DNA double-stranded breaks or single-stranded breaks. Chronic exposure to CdtB can reduce the damage response system and increase the chance of mutations. (**C**) When *Bacteroides fragilis* toxin (BFT) produced by *Bacteroides fragilis* binds to E-cadherin, it can cause cleavage of the protein receptor, Β-catenin, normally bound E-cadherin dissociates and becomes a transcription factor for cell proliferation. BFT also delays the apoptosis of intestinal epithelial cells.

## Data Availability

Not applicable.

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
