# Peer review of "Gut Microbiome in Inflammatory Bowel Disease: Role in Pathogenesis, Dietary Modulation, and Colitis-Associated Colon Cancer"

_microorganisms, 2022, doi:10.3390/microorganisms10071371_

Round 1

Reviewer 1 Report

The manuscript by Gubatan and colleagues review the role of the gut microbiome on the pathogenesis of inflammatory bowel disease (IBD) and colitis-associated cancer (CAC).  The authors discuss the tripartite relationship between the different phyla of bacteria, their bacteriophages, and intestinal epithelial cells. The use of the term “viral dark matter” to describe bacteriophages that are not characterized and/or not culturable was both amusing but appropriate in this instance. The clinical relevance of the gut microbiome  in IBD and the role diet modifying the microbiome in IBD patients will be of interest to the reader. In the final section the authors relate the gut microbiome to the development of CAC. Overall, I found this manuscript to be well written and relevant to those researchers working in this area.  The  three figures are also well done and support the text. I have a few minor comments:

1) Lines 46-48: The sentence “The microbiome of patients with IBD is characterized by bacterial dysbiosis (i.e., an imbalance of pathogenic and commensal bacteria) with a reduced bacterial diversity at active inflammatory states [14, 15], which in each individual patient is evident at different intestinal locations [16],” is a little confusing. 

Should the statement “with a reduced bacterial diversity at active inflammatory states” be either a) “with a reduced bacterial diversity in active inflammatory states” or b) “with a reduced bacterial diversity at active inflammatory sites…….

Line 70: At the beginning of this line, delete “the.”

Lines 85-86: The sentence with: “but they are also able to be sensed directly of 85

both intestinal epithelial cells as well as innate immune cells,”  may read better as “but they are also able to be sensed directly by intestinal epithelial cells and innate immune cells.”

Line 89:  At the end of the sentence, “bacteriophages-” should be changed to “bacteriophage-.”

Line 97: In this paragraph the authors discuss immune cells and epithelial cells but not endothelial cells.  The authors should either provide a few sentences regarding endothelial cells or leave it out (endothelial cells are discussed in the rest of the manuscript).

Lines 113, 159, and 162: The authors use the term “richness.” Please describe what is meant by richness.

Line 195: The authors refer to “DSS-induced recipients.”  Please define “DSS.”

Line 280-281: In the sentence, “TLR4 knockout mice given DSS to induce IBD experienced insufficient epithelial repair. Please change experienced to “had.”

Author Response

Reviewer #1:

The manuscript by Gubatan and colleagues review the role of the gut microbiome on the pathogenesis of inflammatory bowel disease (IBD) and colitis-associated cancer (CAC).  The authors discuss the tripartite relationship between the different phyla of bacteria, their bacteriophages, and intestinal epithelial cells. The use of the term “viral dark matter” to describe bacteriophages that are not characterized and/or not culturable was both amusing but appropriate in this instance. The clinical relevance of the gut microbiome  in IBD and the role diet modifying the microbiome in IBD patients will be of interest to the reader. In the final section the authors relate the gut microbiome to the development of CAC. Overall, I found this manuscript to be well written and relevant to those researchers working in this area.  The  three figures are also well done and support the text. I have a few minor comments:

 RESPONSE: Thank you for the feedback.

  • Lines 46-48: The sentence “The microbiome of patients with IBD is characterized by bacterial dysbiosis (i.e., an imbalance of pathogenic and commensal bacteria) with a reduced bacterial diversity at active inflammatory states [14, 15], which in each individual patient is evident at different intestinal locations [16],” is a little confusing. Should the statement “with a reduced bacterial diversity at active inflammatory states” be either a) “with a reduced bacterial diversity in active inflammatory states” or b) “with a reduced bacterial diversity at active inflammatory sites…….

RESPONSE: We have revised the section to improve clarity: “The microbiome of patients with IBD is characterized by bacterial dysbiosis (i.e., an imbalance of pathogenic and commensal bacteria). Bacterial diversity has been shown to be reduced during active inflammation [14, 15] in IBD. Furthermore, gut microbiome composition has been shown to vary based on location along the gastrointestinal tract [16].”

Line 70: At the beginning of this line, delete “the.”

RESPONSE: “the” has been deleted

Lines 85-86: The sentence with: “but they are also able to be sensed directly of 85 both intestinal epithelial cells as well as innate immune cells,”  may read better as “but they are also able to be sensed directly by intestinal epithelial cells and innate immune cells.”

RESPONSE: We have revised to “but they are also able to be sensed directly by intestinal epithelial cells and innate immune cells.”

Line 89:  At the end of the sentence, “bacteriophages-” should be changed to “bacteriophage-.”

RESPONSE: We have changed to “bacteriophage-.”

Line 97: In this paragraph the authors discuss immune cells and epithelial cells but not endothelial cells.  The authors should either provide a few sentences regarding endothelial cells or leave it out (endothelial cells are discussed in the rest of the manuscript).

RESPONSE: We have removed endothelial cells from the paragraph for better flow: “Taken together, the above-mentioned studies underscore the importance of understanding the direct effects on bacteriophages, not only on bacteria, but also on both immune and epithelial cells.”

Lines 113, 159, and 162: The authors use the term “richness.” Please describe what is meant by richness.

RESPONSE: Thank you for allowing us to clarify. We have now clearly defined diversity and richness in our manuscript: “In IBD pathogenesis, bacterial dysbiosis is characterized by decreased bacterial diversity (measure of the number of species in a community, and a measure of the abundance of each species) and richness (number of species in a community) evident by…”

Line 195: The authors refer to “DSS-induced recipients.”  Please define “DSS.”

RESPONSE: “that fecal microbiota transplantation from donors on a KD confers microbiota benefits and relieves colitis in dextran sulfate sodium (DSS)-induced recipients…”

Line 280-281: In the sentence, “TLR4 knockout mice given DSS to induce IBD experienced insufficient epithelial repair. Please change experienced to “had.”

RESPONSE: We have revised to “TLR4 knockout mice given DSS to induce IBD had insufficient epithelial repair. TLR4 also plays a role in the proliferation of intestinal epithelial cells [93-95].”

Reviewer 2 Report

The manuscript demonstrates gut microbiome in inflammatory bowel disease, role in pathogenesis, dietary modulation, and colitis-associated colon cancer appropriately and concisely, as results of reviewing many literatures.

Major comments

1.      Line 65-67, you should describe exactly whether the intake of prebiotics like nondigestible fibers positively or negatively correlated to circulating serum levels of GM-CSF, IL-6, IL-8, and TNF-α.

2.      good” bacteria in Line 153, probiotic/commensal bacteria in Line 383-384 and probiotic gut bacteria in Line 392 should be unified into “beneficial” bacteria, beneficial/commensal bacteria and beneficial gut bacteria, respectively. The FAO/WHO definition of a probiotic is “live microorganisms which when administered in adequate amounts confer a health benefit on the host”. Probiotics are often not commensal bacteria.

Author Response

Reviewer #2:

Comments and Suggestions for Authors

The manuscript demonstrates gut microbiome in inflammatory bowel disease, role in pathogenesis, dietary modulation, and colitis-associated colon cancer appropriately and concisely, as results of reviewing many literatures.

RESPONSE: Thank you for the feedback.

Major comments

  1. Line 65-67, you should describe exactly whether the intake of prebiotics like nondigestible fibers positively or negatively correlated to circulating serum levels of GM-CSF, IL-6, IL-8, and TNF-α.

RESPONSE: We have clarified the correlations in the manuscript: “Interestingly, intake of prebiotics like nondigestible fibers are positively correlated with circulating serum levels of granulocyte-macrophage colony stimulating factor (GM-CSF) and negatively correlated with interleukin (IL)-6 and  IL-8.”

  1. “good” bacteria in Line 153, probiotic/commensal bacteria in Line 383-384 and probiotic gut bacteria in Line 392 should be unified into “beneficial” bacteria, beneficial/commensal bacteria and beneficial gut bacteria, respectively. The FAO/WHO definition of a probiotic is “live microorganisms which when administered in adequate amounts confer a health benefit on the host”. Probiotics are often not commensal bacteria.

RESPONSE: Thank you for this correction. We have made the recommended revisions:

“if one should help facilitate the colonization of microbiota by using bowel lavage or antibiotics risking elimination of preexisting beneficial bacteria prior to the therapeutic intervention.”

“IBD is characterized by gut microbial dysbiosis with decreased beneficial/commensal bacteria (Firmicutes, Actinobacteria, and Bacteroides)…”

“Dietary interventions, such as the CDED, ketogenic diet, low-carbohydrate diet, and plant-based diets, may have beneficial effects on patients with IBD through improving gut microbial diversity and increasing beneficial gut bacteria.”
